# Toward Faithfulness-guided Ensemble Interpretation of Neural Network

## Abstract

Interpretable and faithful explanations for specific neural inferences are essential for understanding and evaluating the behavior of models. For this purpose, feature attributions are highly favored for their interpretability. To objectively quantify the faithfulness of an attribution to the model, a widely used metric uses perturbations of the input that mask either the highly salient or highly non-salient features. These metrics, however, neglect the faithfulness of the attribution to the hidden-layer encodings of the model, and hence ignore its internal structure. In response, we propose a novel attribution method, **FEI**, which targets faithfulness to hidden layer representations. Moreover, the method optimizes the quality of the attribution according to the perturbation metrics using a novel smooth approximation of the metrics that allows effective optimization by gradient decent. This improve its performance on faithfullness evaluation. The method provides enhanced qualitative interpretability, while also achieving superior scores in quantitative faithfulness measurements.

## 1 Introduction

The complex structure and high-dimensional embedding of state-of-the-art Neural Networks render them essentially uninterpretable to humans. This presents a significant barrier to comprehending their internal functioning. It has been established that a good explanation should satisfy two criteria: faithfulness and interpretability Jacovi & Goldberg (2020); Selvaraju et al. (2017). Faithfulness measures the extent to which the explanation accounts for the reasoning process of the model, while interpretability emphasizes the degree to which the explanation is comprehensible to humans. These two criteria are mutually exclusive and can be evaluated separately. Much of the recent research on explaining neural networks focuses on generating attribution maps for input images Sundararajan et al. (2017); Smilkov et al. (2017); Chattopadhay et al. (2018),as these maps offer substantial interpretability with their visualizations.. An attribution map consists of importance values that quantitatively measure the causal relationship with the neural network output for a particular input. In other words, it assigns a heatmap to the input based on its relevance to model inference.

Evaluating attribution presents challenges since there is no universal agreement on objective metrics for assessing the quality of explanations. Interpretability of explanation is often evaluated qualitatively through visualization or quantitatively through human rating scores Ribeiro et al. (2016); Selvaraju et al. (2017). However, measuring improvements in interpretability is challenging because it often involves human subjects. Consequently, researchers have shifted their focus to enhancing faithfulness. A new category of explanation methods has emerged, driven by faithfulness evaluation. These methods observe the network's output with perturbed input, typically investigating changes in the output when specific input features are removed. The underlying intuition behind these techniques is that explanation in the form of attribution map is faithful if highly-attributed features alone can produce output similar to the full input. Nevertheless, existing methods Fong & Vedaldi (2017); Wagner et al. (2019) depend on a coarse approximation of faithfulness metrics as their objectives. This approach demands significant effort in hyperparameter tuning and still yields suboptimal results. Thus, a better objective is required for improve overall faithfulness.

Moreover, prior research often overlooks the intermediate layers of neural networks when assessing faithfulness. While neural networks are frequently perceived as black-box models, they can be regarded as white-box models because every computation they perform and every value they compute

in the model are visible to users. However, what renders these models intrinsically uninterpretable is the sheer magnitude of these variables, often numbering in the millions or even billions, making them impossible for human to comprehend. The white-box nature of neural networks implies that the model's reasoning process is encoded in the intermediate part of the model. Since faithfulness requires that the explanation accurately reflects the model's reasoning process, it is reasonable to assert that faithfulness measurements should encompass the hidden part of the model. In this paper, we focus on addressing the aforementioned challenges of both overall and internal faithfulness.

First, we introduce a novel ensemble method that closely aligns with perturbation metrics. Instead of optimizing a single attribution map, this approach formulates a superior approximation by optimizing multiple attributions, thereby achieving greater similarity with the actual evaluation metrics. Additionally, through the use of the ensemble method, we are able to eliminate the need for hyper-parameters. Further details can be found in Section 3.1.

Furthermore, we achieve internal faithfulness by implementing gradient clipping techniques within the hidden layers. We choose gradient clipping due to its lightweight nature and the absence of additional gradient introductions. Additionally, working with gradients from perturbation optimization helps maintain overall faithfulness. We provide implicit evidence of the effectiveness of our technique in preserving faithfulness within the hidden layers through qualitative measurements via visualization, as demonstrated in section 4.3. Our regulation method also prevents the emergence of adversarial noise during optimization, resulting in interpretable visualizations and improved quantitative faithfulness scores.

In this work, our primary focus lies within the domain of computer vision. However, we firmly believe that our method can be extended to other domains. In the evaluation section, we demonstrate the superiority of our techniques by achieving higher scores in quantitative faithfulness evaluations and by providing accurate and interpretable visualizations.

## 2 BACKGROUND

### 2.1 EXPLANATION GENRE

There is a wide range of prior work focus on explanation, mainly feature attribution.

**Backpropagation-based Methods** These methods calculate importance scores for input images by back-propagating information to the input . Such approaches are efficient as they typically require only a single forward and backward pass. For instance, Simonyan et al. (2013) generate a saliency map based on ordinary gradients, while other works, including Zeiler & Fergus (2014); Smilkov et al. (2017); Wagner et al. (2019); Springenberg et al. (2014) regulate the gradient to reduce noise. . Alternatively, some methods propagate different measurements based on gradients or activations to the input layer, as seen in works by Shrikumar et al. (2017); Sundararajan et al. (2017); Bach et al. (2015), While these attribution maps are typically fine-grained, they can be challenging to interpret.

**Activation-based Methods** Activation-based methods aggregate the values of a specific layer to create a heatmap for input images Zhou et al. (2016); Selvaraju et al. (2017); Chattopadhay et al. (2018). While these visualizations are generally at a coarse level, they can be combined with back-propagation methods such as those in Springenberg et al. (2014) to produce fine-grained visualizations at the input. However, it's important to note that these approaches can sometimes yield unfaithful results, as demonstrated by Adebayo et al. (2018).

**Surrogate Methods** Another approach to explaining neural network decisions is to create a simple and interpretable surrogate model based on input and output examples. For instance, distillation Hinton et al. (2015) a shallow network from a deep network. LIME Ribeiro et al. (2016) , on the other hand, creates a linearized approximation of the target network's behavior around a given input image.

**Perturbation-based Methods** Perturbation-based methods revolve around the concept of perturbing individual inputs or neurons and examining the network's output. Some approaches, such as those in Petsiuk et al. (2018) and Sun et al. (2020), involve masking parts of images to calculate importance scores based on changes in model output. These methods vary in their techniques for occlusion and the heuristics used for ranking image features. Another category of perturbation-

based methods focuses on optimization, seeking to find a mask that has the maximum or minimum impact on the model's output Fong & Vedaldi (2017); Fong et al. (2019); Wagner et al. (2019). However, the optimization process often introduces adversarial noise, necessitating various regulation methods such as mask smoothing Fong et al. (2019), gradient clipping Wagner et al. (2019) and resolution reduction Fong & Vedaldi (2017). While these methods are typically faithful to the model's predictions by design, they can be relatively computationally inefficient. Our method falls into this category. By defining a more appropriate objective and promoting internal faithfulness, our method enhances evaluation scores while simultaneously improving efficiency.

## 2.2 EVALUATION FOR EXPLANATION

Different evaluation mechanisms for explanations have been proposed to assess their quality. Most of these mechanisms can be divided into two categories: Interpretability and Faithfulness.

**Intepretability** evaluation often requires explicit or implicit human judgment of the explanation. Experiments are conducted by hiring users to rate the quality based on its interpretability, as seen in previous studies Lundberg & Lee (2017) or its effectiveness in aiding specific tasks, as demonstrated by Ribeiro et al. (2016). Interpretability can also be quantitatively measured with implicit human judgment using methods such as the pointing game Zhang et al. (2018), which assesses the quality of the explanation by evaluating its alignment with a human-annotated segmentation map.

**Faithfulness** evaluation often involves measuring the relationship between the explanation and the model. One prevalent faithfulness test is the sanity test Adebayo et al. (2018), which evaluates faithfulness to the model by comparing the change in visualization with parts of the model weight randomized. Another widely used metrics are the MoRF and LeRF tests proposed by Samek et al. (2016), also known as the deletion and preservation tests. In essence, the deletion test posits that the removal of pixel information, starting from the most to the least salient pixels, should lead to a rapid decrease in the model output. On the contrary, preservation test remove pixel from the least to the most salient, resulting in a slow decrease in the category output. The test results can be quantitatively measured by plotting the curve of model output versus the deleted/preserved area fractile and computing the Area Under Curve Binder et al. (2016). Some variations of this metric have been proposed. While most variations introduce changes in the perturbation Kapishnikov et al. (2019), the remove-and-retrain metrics measure the change in model accuracy when retrained with perturbed data Hooker et al. (2019). Our method is guided by existing faithfulness evaluation metrics while exploring faithfulness within the internal structure of neural networks.

## 3 METHOD

We begin by establishing a clear definition of faithfulness. Intuitively, an attribution is considered faithful when, for a given input value, features with a higher influence on the output possess correspondingly higher saliency values. By influence, we mean a tendency to reinforce the probability of the top category. One way to evaluate the influence of an input feature involves masking it in some manner and observing the resultant effect on the output. When masking a highly salient feature, we expect a reduction in the top category output, whereas masking a feature of lower salience should generally preserve it. However, when we consider a feature to be a single pixel, masking it often fails to have a meaningful impact on the category output. Thus, we must shift our focus towards assessing the influence of *ensembles* of pixels. We define an attribution as faithful if ensembles of input features with greater influence exhibit higher aggregate salience.

To formalize this notion, we introduce a perturbation operator $\otimes$. Given an input value $x$ (typically an image) and an ensemble of input features $S$, $x \otimes S$ yields a perturbed input value. This perturbation is usually achieved by masking the features within the ensemble $S$. We consider perturbation operators that mask pixels by replacing them either with a fixed background value or with a random value. Define $R = (x \otimes P)$ as a fully masked reference image, where $P$ is the ensemble of all pixels. Let $\phi$ be the function that yields the top category output of a neural network on input $x$. We define the *influence* of ensemble $S$ as $\phi(x) - \phi(x \otimes S)$, that is, the reduction in the estimated category probability resulting from masking the features within the ensemble $S$.

## 3.1 FAITHFULNESS METRICS

Suppose we are given an attribution map $M$, which assigns real numbers to features indicating their salience. To assess the faithfulness of $M$, we need to decide which ensembles of features we should compare. For this purpose, Binder et al. (2016) propose to consider the *fractiles* of $M$. Given a fraction $f \in [0, 1]$, the *upper $f$-fractile* is defined as the set of pixels with saliency values greater than a fraction $f$ of all the pixels. Conversely, the *lower $f$-fractile* consists of pixels with saliency values *less* than a fraction $f$ of all the pixels. Intuitively if the attribution is faithful, the upper fractiles should exhibit relatively high influence, while the lower fractiles should exhibit relatively low influence. Formally, we define a *preservation metric* which varies inversely with the average influence of the lower fractiles, that is:

$$Q_P = \langle \phi(x \otimes l_f(M)) \rangle_{f \in \mathcal{F}} \tag{1}$$

Here, $l_f(M)$ is the *lower $f$-fractile* of $M$ and $\mathcal{F}$ is the set of fractions $\{1/i \mid i = 0 \dots N\}$ for $N$ the number of features. This measure will be high if the category output decreases only slightly as we mask pixels from least to most salient. Similarly, the *deletion metric* is defined in terms of the upper fractiles $u_f(M)$, that is, $Q_D = \langle \phi(x \otimes u_f(M)) \rangle_{f \in \mathcal{F}}$. For this metric, lower is better. It is *low* if the category output decreases rapidly as pixels are masked from the most to least salient.

## 3.2 FAITHFULNESS OPTIMIZATION

Fong & Vedaldi (2017) introduced a new category of *perturbation methods* that optimize the attribution using a proxy for the perturbation metrics as the objective. This presents a challenge, as the fractiles are not differentiable functions of the attribution $M$. To address this, perturbation methods typically define a smoothed objective function in terms of a single perturbed image $\tilde{x}$. This perturbed image is a linear interpolation between the original image and the fully masked image $R$. That is, $\tilde{x} = M \cdot x + (1 - M) \cdot R$. The approximated preservation objective is to maximize $\phi(\tilde{x})$.

The objective mentioned above, while differentiable in terms of the attribution map $M$, presents a coarse approximation due to its consideration of only one perturbed image, as opposed to a sequence of $N$ perturbed images. Moreover, it tends to have a trivial maximum at $M = 1$ which is clearly not faithful. To mitigate this issue, a regularization term is often introduced to impose an area constraint on the attribution map the $M$. For example, in Fong et al. (2019), a ranking-based area loss is introduced that forces the non-zero part of the attribution to have a fixed area $\beta$. However, setting the hyperparameter $\beta$ in a principled manner can be challenging. Lower area constraints often do not include sufficient information for adequate explanations, while higher values tend to produce noticeable artifacts.

To address this issue, we propose to approximate the perturbation metrics more accurately using a smooth approximation of the fractiles of $M$. This eliminates the need for the area hyperparameter. Consider optimizing the preservation metric, where our goal is to maximize the category output for the lower fractiles of $M$. For a given fraction $f$, we approximate the lower $f$-fractile by a mapping $\alpha_f$ from pixels to the interval $[0, 1]$. We chose not to employ a binary mapping because it is computationally expensive to optimize and often results in suboptimal solutions. We interpret $\alpha_f(p)$ as the probability that pixel $p$ is *not* in the lower $f$-fractile of $M$. The map must satisfy the constraint $\Sigma_p \alpha_f(p) = (1 - f) \cdot N$, where $N$ is the number of pixels. For each fractile, we can approximate the expected values of the perturbed image as follows:

$$\tilde{x} = \alpha_f \cdot x + (1 - \alpha_f) \cdot R \tag{2}$$

In the case where $\alpha_f$ is binary, this approximation is exact. We can minimize the influence of lower fractile $f$ by maximizing $\phi(\tilde{x})$ subject to the constraint $\Sigma_p \alpha(p) = (1 - f) \cdot N$. In practice, we soften this constraint by including a loss term proportional to the area error. For the preservation metric, our loss function becomes:

$$l_{faith} = -\phi(\tilde{x}) + \beta_1 |\Sigma_p \alpha_p - (1 - f) \cdot N| \tag{3}$$

where $\beta_1$ is a hyperparameter which has minor effect on the optimization since we always expect the constrained to be satisfied. Now consider the entire sequence of fractiles $\{\alpha_f\}$ for $f$ in $\mathcal{F}$. We say this sequence is *consistent* if $\alpha_a \leq \alpha_b$ for $a \geq b$. That is, if a pixel is in the lower 0.1-fractile, it must be in the lower 0.2-fractile, and so on. If the sequence of fractiles $\{\alpha_f\}$ is consistent

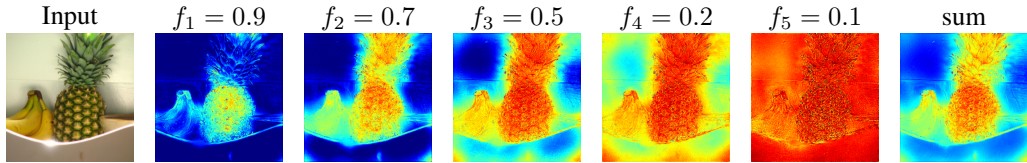

Figure 1: attribution map generated by $\text{FEI}_{\text{IBM}}$ for pineapple at different fractile

and binary, we can reconstruct an attribution $M$ from these fractiles by simply summing them as $M = \Sigma_{f \in \mathcal{F}} \alpha_f$. However, in practice, simultaneously optimizing all of the fractiles under these constraints is challenging. Instead, we choose a small set of fractions $f_1 \ldots f_k$ and optimize each $\alpha_{f_i}$ independently using gradient descent. The resulting saliency map is given by:

$$M = \Sigma_{i=1}^k \alpha_{f_i} \tag{4}$$

In effect, since the fractiles $\alpha_{f_i}$ are smooth, the missing fractiles are obtained by interpolating between them. Moreover, to encourage consistency of the fractiles, we optimize them in a cascading manner, using $\alpha_{f_i}$ as the initial value for $\alpha_{f_{i+1}}$. This set up also enables us to reduce the number of iterations during the optimization to save computation cost. To optimize for the deletion test, we simply invert the sense of $\alpha_f$ in Equation. 2 above, giving us:

$$\tilde{x} = (1 - \alpha_f) \cdot x + \alpha_f \cdot R$$

and we minimize rather than maximize $\phi(\tilde{x})$. We will refer to this optimization approach as the *ensemble method*. Experimentally, we find that this approximation of the perturbation metrics gives good results, and that we can effectively optimize it by gradient descent. A visualization of the ensemble method is shown in figure 1.

## 3.3 INTERNAL FAITHFULNESS

Independently optimizing the fractiles of the attribution map without considering their consistency offers the advantage of computational tractability. However, it also introduces a significant risk of over-optimization of the fraUnderctiles. While the intention is to interpret masking pixels as the elimination of information from the input image, it can inadvertently perturb the image in a way that activates feature recognizers in the hidden layer that are not directly relevant to the classification of the input image. This phenomenon, known as 'adversarial noise,' has been observed in prior work Ghorbani et al. (2019). Various regularization methods have been proposed to address this challenge Du et al. (2018); Wagner et al. (2019); Fong et al. (2019). We contend that this problem arises due to a lack of faithfulness and can be mitigated through the achievement of *Internal Faithfulness*.

Conceptually, *Internal Faithfulness* implies that the explanation should mirror a reasoning process similar to the original model's inference. When optimizing for explanations, it's possible to adopt a different reasoning process, which can lead to the generation of artificial features in the explanation. Therefore, striving for internal faithfulness is essential to tackle this problem. However, understanding the precise reasoning process of model inference is challenging, given that the latent encoding of a neural network is not interpretable to us. To attain internal faithfulness, we must identify a suitable proxy that effectively represents the model's reasoning process.

An intuitive approach is to align the internal activation map with the original inference process during optimization. However, blindly matching activations may be detrimental, as it may prioritize the less important parts while neglecting the truly essential ones. In contrast, to prevent the activation of internal feature detectors that were not activated in the unmasked image, Wagner et al. (2019) proposed an internal regulation method that clips gradient computations during optimization if the activation values fall outside a certain feasible range. As demonstrated in Section 4.2, this method exhibits sub-optimal performance in evaluation metrics due to its lack of flexibility. We believe we can adopt a similar clipping technique to achieve internal faithfulness.

There are several advantages to using gradient clipping techniques. Firstly, it is applied directly to gradients with respect to the loss function in equation 3, which contribute to enhanced overall faithfulness. Leveraging these signals for activation matching can potentially dynamically align

activations that are intrinsically faithful. Additionally, if we were to introduce a new regulation term, additional gradients might compromise the overall faithfulness gradient, potentially leading to suboptimal solutions. Moreover, gradient clipping is a relatively cost-effective option, as it does not require the computation of new gradients for every neuron in the hidden layers.

Formally, suppose the model $\phi(x)$ can be decomposed into $\rho \circ \psi$, where $\psi$ is the function of a hidden layer with respect to the inputs, we have $x^h = \psi(x) \in \mathbb{R}^{C^h \times H^h \times W^h}$ as the hidden layer output at layer $h$. During the optimization for fractile $f$, the activations of hidden layer with perturbed input can be written as $\tilde{x^h} = \psi(\tilde{x})$. let $\gamma_h$ be the gradient of the category output with respect to the perturbed activation at the hidden layer, that is, $\gamma_h = \dfrac{\partial \psi(\tilde{x}^h)}{\partial \tilde{x}^h}$ and $\tilde{\gamma_h}$ be the clipped gradient.

A trade-off exists between the internal and overall faithfulness metrics when employing the gradient clipping method. Stronger gradient clipping reduces overall faithfulness because it clips more gradients during optimization. To thoroughly explore this trade-off, we introduce four distinct gradient clipping methods, each characterized by varying levels of strength. To start with, we introduce **V**alue **M**atching (**VM**)

$$
\tilde{\gamma_h} = \begin{cases} 0 & \text{if } \gamma_h \geq 0 \wedge \bar{x}^h > x^h \\ \gamma_h & \text{if } \gamma_h \geq 0 \wedge \bar{x}^h \leq x^h \\ \gamma_h & \text{if } \gamma_h < 0 \wedge \bar{x}^h > x^h \\ 0 & \text{if } \gamma_h < 0 \wedge \bar{x}^h \leq x^h \end{cases} \tag{5}
$$

The derivation of this equation can be found in appendix. This clipping mechanism aims to align with the exact values of the unperturbed hidden layer. When the activation with perturbed input exceeds that of the unperturbed one, we clip the positive gradient, preventing it from increasing further, implicitly causing it to decrease. Conversely, when the activation with masked input is lower than the unmasked one, we restrain the gradient that decreases the activation. This gradient clipping strategy match the exact values of activation with unperturbed input at the cost of clipping many gradients beneficial for overall faithfulness. To alleviate the strength of gradient clipping, we introduce a decomposition of the VM method into two new gradient clipping methods as follow:

$$
\tilde{\gamma_h} = \begin{cases} 0 & \text{if } \gamma_h \geq 0 \wedge \tilde{x}^h \leq x^h \\ \gamma_h & \text{Otherwise} \end{cases} \tag{6}
$$

$$
\tilde{\gamma_h} = \begin{cases} 0 & \text{if } \gamma_h < 0 \wedge \tilde{x}^h > x^h \\ \gamma_h & \text{Otherwise} \end{cases} \tag{7}
$$

In equation 6, we encourage the reduction of overly activated neurons by clipping any gradient that attempts to increase their activation. In the case of asymmetric activation function such as ReLU, this method encourages the inactivated part with unperturbed input to remain inactive, and as such, we term it **I**nactivated **V**alue **M**atching(**IVM**). Similarly, in Equation 7, we encourage the activated part to remain active, which can be termed **A**ctivated **V**alue **M**atching(**AVM**). Furthermore, when asymmetric activation functions are involved, the gradient clipping condition can be further relaxed, leading to what we call **I**nactivated **B**inary **M**atching (**IBM**).

$$
\tilde{\gamma_h} = \begin{cases} 0 & \text{if } \gamma_h \geq 0 \wedge x^h \leq 0 \\ \gamma_h & \text{Otherwise} \end{cases} \tag{8}
$$

In this case, our objective is to match the binary status of $x^h$. Equation 8 is applied by by introducing a condition $x^h \leq 0$ to equation 6 This condition ensures that we only clip the gradient for the original inactivated neurons while disregarding the value of the activated ones. Similarly, a condition of $x^h \geq 0$ can be added to equation 7, but it doesn't alter the clipping condition.

With four distinct gradient clipping methods, we are able to achieve internal faithufulness while maintaining overall faithfulness. In practical implementation, we only apply those gradient clipping methods to feature extraction layers. With those gradient clipping methods and our ensemble technique, we proudly present **F**aithfulness-guided **E**nsemble **I**nterpretation, referred to as FEI. This comprehensive approach encompasses four variations, denoted as $\text{FEI}_{\text{VM}}$, $\text{FEI}_{\text{IVM}}$, $\text{FEI}_{\text{AVM}}$, and $\text{FEI}_{\text{IBM}}$, each tailored to specific faithfulness considerations.

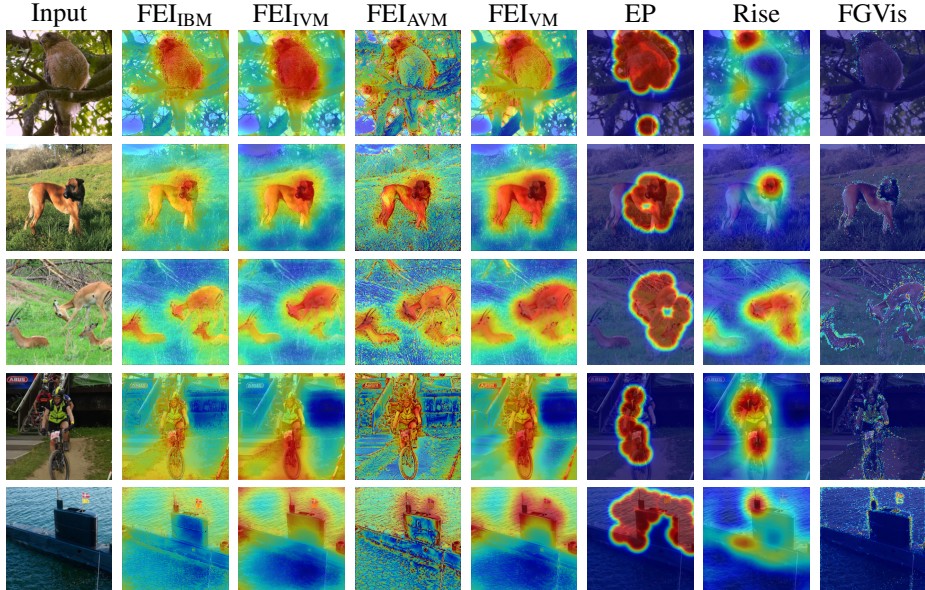

| Input | FEI$_{IBM}$ | FEI$_{IVM}$ | FEI$_{AVM}$ | FEI$_{VM}$ | EP | Rise | FGVis |

Figure 2: We compared our method with to other perturbation methods. Results generated from its official implementation and our re-implementation. The attribution are generated for VGG16 Simonyan & Zisserman (2014) network Goodfellow et al. (2016). The object in top down order are *Bald Eagle*, *Boxer*, *Impala*, *Mountain bike*, and *Submarine*

## 4 EVALUATION

We now apply the method to obtain insight into the functioning of CNN's on image recognition tasks. All evaluation metric use the ImageNet validation datasetRussakovsky et al. (2015). Reference image with random monotone color is used in each optimization iteration and the ensemble of fraction $f_k \in [0.1, 0.3, 0.5, 0.7, 0.9]$ are applied. Map in each fratiles is optimized for 100 iteration with $\beta_1 = i \cdot e^{-2}$ which increase with the epoch Numbers $i$. Each map is optimized for 100 iterations with Adam optimizer Kingma & Ba (2014). The methods we are comparing against includes Extremal Perturbation(EP) Fong et al. (2019), RISE Petsiuk et al. (2018), FGVis Wagner et al. (2019), GradCam Selvaraju et al. (2017) and Integrated Gradient Sundararajan et al. (2017).

### 4.1 VISUALIZATION

A qualitative visualization comparison among methods is presented in Figure 2. In comparison to coarse attribution maps, our method demonstrates a more precise identification of the subject. In contrast to other fine-grained attribution maps, our visualization conveys more information about the overall image, as opposed to merely displaying isolated edge maps. Subjectively, our approach provides superior visualizations. Among variants of our methods, FEI$_{AVM}$ seems to focus on matching edge maps, therefore producing a more noisy map.

### 4.2 PERTURBATION RESULT

Table 1 presents the preservation and deletion scores, comparing our method with prior methods. Each method is evaluated using 1000 images, and results for stochastic methods are reported as an average of three measurements. We plot the softmax score of the desired category against the area threshold and compute the AUC for both metrics. Higher values are better for preservation, while lower values are better for deletion. FEI$_{IBM}$ achieves the best score in the preservation metric, while Integrated Gradient performs best in the deletion score. However, we still obtain a better deletion score than other methods. One reason for FEI$_{IBM}$ performing best among the clipping methods is that it has the weakest regulation.

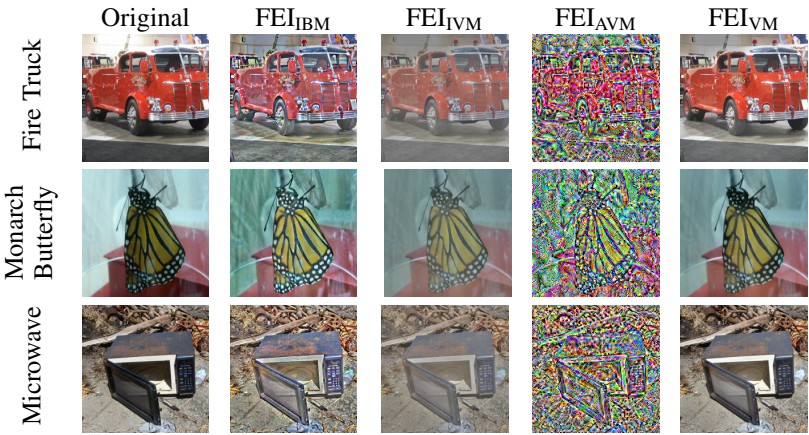

Figure 3: Image reconstruction from gradient clipping. The model we use is VGG16

Table 1: Quantitative Faithfullness evaluation of perturbation method with Gaussian-blurred image as reference for preservation and Grey image as reference for deletion on VGG16 Simonyan & Zisserman (2014) and ResNet34 He et al. (2016), Implementation from RISE Petsiuk et al. (2018)

| Method | VGG16 | | ResNet34 | |
|---|---|---|---|---|
| | Preservation | Deletion | Preservation | Deletion |
| GradCam | 0.624 | 0.128 | 0.648 | 0.138 |
| Integrated Gradient | 0.362 | **0.041** | 0.516 | **0.065** |
| RISE | 0.620±0.014 | 0.118±0.0048 | 0.642±0.007 | 0.129±0.005 |
| Extremal Perturbation | 0.461±0.010 | 0.236±0.001 | 0.520±0.008 | 0.234±0.009 |
| FGVis | 0.400±0.011 | 0.060±0.001 | 0.469±0.005 | 0.132±0.006 |
| $FEI_{IBM}$ | **0.653**±0.005 | 0.079±0.003 | **0.722**±0.007 | 0.139±0.004 |
| $FEI_{IVM}$ | 0.582±0.008 | 0.154±0.006 | 0.643±0.005 | 0.094±0.001 |
| $FEI_{AVM}$ | 0.620±0.004 | 0.126±0.006 | 0.671±0.012 | 0.116±0.004 |
| $FEI_{VM}$ | 0.601±0.007 | 0.231±0.005 | 0.647±0.009 | 0.095±0.003 |
| $FEI_{no-clipping}$ | 0.496 ±0.003 | 0.250±0.005 | 0.575±0.005 | 0.351±0.005 |
| $IBM_{no\ ensemble}$ | 0.460 ±0.001 | 0.075±0.003 | 0.650±0.002 | 0.136±0.004 |
| $IBM_{L_1}$ | 0.602 ±0.007 | 0.085±0.002 | 0.638 ±0.009 | 0.096±0.0027 |

### 4.2.1 ABLATION STUDY

In our ablation study, we aimed to assess the effectiveness of our method using $FEI_{IBM}$ as the baseline. We implemented several variations for comparison: $FEI_{no-clipping}$: This implementation employed the ensemble method without the gradient clipping regulation; $IBM_{no\ ensemble}$: In this variant, we employed gradient clipping but used a coarse approximation similar to the method proposed in Fong & Vedaldi (2017);$IBM_{L_1}$: We replaced the ensemble method with an $L_1$ regulation and manually tuned the hyperparameters to achieve the best results. Remarkably, all three of these implementations yielded inferior results compared to our method. This suggests the superiority of our approach. Additional ablation study for design choices can be found in the appendix.

### 4.3 IMAGE RECONSTRUCTION

We propose an image reconstruction metric to evaluate the effectiveness of internal faithfulness. The intuition is that similar internal representations during inference should imply a similar input. We attempt to regenerate an image from scratch using optimization with an objective function to maximize the category output. The results with different gradient clipping techniques are shown in Figure 3. The optimized image are clipped to be within the feasible range of images. Surprisingly, gradient clipping techniques alone are sufficient to recover the image to a strong extent. We can see

| Input | Original | Linear.6 | Linear.3 | Linear.0 | Feature.28 | Feature.26 | Feature.24 | Feature.22 |

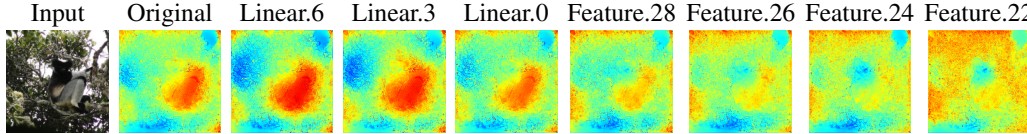

Figure 4: Sanity Check for *Indri* with VGG16 and FEI$_{\text{IBM}}$

Table 2: How often a explanation toward the adversarial target. Results are generated on Alexnet Krizhevsky et al. (2012),ResNet50 He et al. (2016),VGG16 Simonyan & Zisserman (2014) and GoogLeNet Szegedy et al. (2015)

| Model | Defense mechanism | | | | | |
| | FGiVs | FEI$_{\text{IBM}}$ | FEI$_{\text{IVM}}$ | FEI$_{\text{AVM}}$ | FEI$_{\text{VM}}$ | No Clipping |
| --- | --- | --- | --- | --- | --- | --- |
| Alexnet | 0.2% | 4.2% | 1.3% | 95.7% | 0.4% | 100% |
| ResNet50 | 0.2% | 0.1% | 0% | 0% | 0.1% | 100% |
| VGG16 | 0% | 0.1% | 0.1% | 96.0% | 0.1% | 100% |
| GoogLeNet | 0.1% | 0.3% | 0% | 0.1% | 0% | 100% |

that FEI$_{\text{VM}}$ produces a reconstruction with the best quality, while FEI$_{\text{IBM}}$ and FEI$_{\text{IVM}}$ present similar results. FEI$_{\text{AVM}}$ shows noisy images, which corresponds to the result of the defense experiment.

### 4.4 DEFENSE EVALUATION

The proposed gradient clipping methods help the explanation achieve internal faithfulness during optimization. As a result, they also enhance the defense against adversarial noise, which is common in perturbation-based methods. We conducted a quantitative evaluation of the defense effects of our methods, as summarized in Table 2. We tested the frequency of successfully generating an attribution map for a black image across all targets, following the experimental setup of *FGVis* (Wagner et al., 2019), with which we compare our results. Our gradient clipping methods show outstanding ability in preventing adversarial attributions, with the exception of FEI$_{\text{AVM}}$ on AlexNet and VGG16. Its inferior results may be due to the fact that it is matching the activated unit instead of the inactivated unit. Therefore, we can conclude that for neural networks with simple structures, matching the inactivated unit is essential in preventing adversarial noise.

### 4.5 SANITY CHECK

The sanity check proposed by Adebayo et al. (2018) is used to examine the change in attribution when model weights are randomized. We adopted the cascading random setup and visualized the maps using our techniques. The results show more significant and messier changes but with some preserved structure. A more substantial change occurs when we start randomizing feature layers, as our gradient clipping methods are applied in those layers.Some structure is preserved as the internal faithfulness regulation encourages faithfulness in the unrandomized layers. Furthermore, as shown in the defense experiment, the gradient clipping regulation may prevent optimization from finding a feasible map when the generated map is not faithful to the model.

## 5 CONCLUSION

We have introduced **F**aithfulness-guided **E**nsemble **I**nterpretation, a novel neural network explanation method driven by faithfulness in both model decisions and internal model functioning. Our approach combines ensemble approximation, inspired by quantitative faithfulness metrics, with gradient clipping, inspired by internal activation matching. Additionally, we've proposed a new qualitative metric that implicitly assesses internal faithfulness. FEI has demonstrated superior results in both qualitative visualization and quantitative evaluation. In the future, our goal is to develop a direct quantitative measure for assessing internal faithfulness and an objective measure for interpretability. We plan to enhance our method based on these metrics.

## 6 ETHICAL STATEMENT

Our paper enhances the understanding of neural network functioning, which potentially aids in attacks that can fool neural network predictions. However, we believe that overall neural network explanations will help identify bias in neural networks and better serve society. In future research, we will also replace the original ImageNet dataset Russakovsky et al. (2015) with the face-obscured version Yang et al. (2021).

## 7 REPRODUCIBILITY

The code we used to produce our results is included in the supplementary material. Unfortunately, we did not seed our results, so reproduction may not be exact. However, using the provided code, one should be able to reproduce results that are close enough to identify the effectiveness of our method.

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

## A    APPENDIX

### A.1    DERIVATIONS

We present a derivation of equation 5.

Since we want to match the internal structure, We define a distance function $d(x, y) = \|x - y\|$ to measure the difference of hidden layer activation's. However, As we discussed, not all activation are relevant to the model decision. Therefore, we assume a saliency map $M^h$ in hidden layer to guide the regulation. A new approximated loss term can be added for internal faithfulness:

$$l_{int} = \epsilon \sum_{j \in \sigma} M_j^h d(x_j^h, \tilde{x}_j^h) = \epsilon \sum_{j \in \sigma} M_j^h |x_j^h - \tilde{x}_j^h| \tag{9}$$

where $\epsilon$ is a hyperparameter. If we know the hidden layer saliency map $M^h$, then we can simply add this loss term to the optimization objective for each $\alpha_f$ and thus balance the optimization of the internal and external optimization metrics using the hyperparameter $\epsilon$.

However, $M^h$ remains unknown to us. As a solution, we propose a simple approximation using $\gamma_h = \dfrac{\partial \psi(\tilde{x}^h)}{\partial \tilde{x}^h}$. We have noticed that the gradient is commonly used as a measure of saliency. Therefore, we can consider the gradient as an approximation of the hidden layer saliency map $M^h$. Although the gradient can be either positive or negative, our primary concern is its relevance to the output. Thus, we can approximate $M^h$ as $|\gamma_h|$. With this approximation in mind, the gradient of the loss function with respect to $\tilde{x}^h$ can be written as:

$$\frac{\partial l_{int}}{\partial \tilde{x}^h} \approx \epsilon \begin{cases} \gamma_h & \text{if } \gamma_h \geq 0 \wedge \tilde{x}^h > x^h \\ -\gamma_h & \text{if } \gamma_h \geq 0 \wedge \tilde{x}^h \leq x^h \\ -\gamma_h & \text{if } \gamma_h < 0 \wedge \tilde{x}^h > x^h \\ \gamma_h & \text{if } \gamma_h < 0 \wedge \tilde{x}^h \leq x^h \end{cases} \tag{10}$$

Now let us consider the gradient for the sum of loss for the external and internal faithfulness metrics. The gradient external preservation loss just $\dfrac{\partial l_{faith}}{\partial \tilde{x}^h} = -\gamma_h$. Summing this two gradients with $\epsilon = 1$, we have

$$\frac{\partial l_{faith} + \partial l_{int}}{\partial \tilde{x}^h} \approx \begin{cases} 0 & \text{if } \gamma_h \geq 0 \wedge \tilde{x}^h > x^h \\ -\gamma_h & \text{if } \gamma_h \geq 0 \wedge \tilde{x}^h \leq x^h \\ -\gamma_h & \text{if } \gamma_h < 0 \wedge \tilde{x}^h > x^h \\ 0 & \text{if } \gamma_h < 0 \wedge \tilde{x}^h \leq x^h \end{cases} \tag{11}$$

$\epsilon = 1$ implies the internal and external losses are weighted equally. We ignore the constant as it can be balanced with learning rate.

Table 3: Ablation result for approximation choice

|  | Preservation | Deletion |
| --- | --- | --- |
| $\text{FEI}_{\text{IBM}}$ (Independent training) | 0.646±0.005 | 0.078±0.003 |
| $\text{FEI}_{\text{IBM}}$ (Binary Map) | 0.631±0.006 | 0.091±0.004 |
| $\text{FEI}_{\text{IBM}}$ (fully trained) | 0.660±0.00 | 0.1088±0.004 |
| $\text{FEI}_{\text{IBM}}$ | **0.653**±0.005 | 0.079±0.003 |

## A.2 ADDTIONAL ABALTION STUDY

We examined the design approximations we made to assess their effectiveness as shown in table 3, with $\text{FEI}_{\text{IBM}}$ as the baseline on the VGG16 network. We implemented several variations for comparison. In $\text{FEI}_{\text{IBM}}$ (Independent training), we trained from scratch for each fractile instead of training them in a cascading order. In $\text{FEI}_{\text{IBM}}$ (Binary Map), we added a regularization term to force the map to be binary.

$$-\phi(\tilde{x}) + \beta_1 |\Sigma_p \alpha_p - (1 - f) \cdot N| + \beta_2 |\alpha_p \cdot (1 - \alpha_p)| \tag{12}$$

We manually tuned the value of $\beta_2$ to optimize it and increased the training time to 500 iterations. With $\text{FEI}_{\text{IBM}}$ (fully trained), we extended the training time to 500 epochs for each iteration for comprehensive training. As shown in the results, the cascading training method proved effective in improving the results. Although the binary map provides an exact approximation for each fractile, finding an optimal point with regularization is challenging, leading to suboptimal results and longer training times. Ultimately, increasing the training time can be helpful in achieving better results, but it might not be worth the cost given its expense.

