# OpenReview forum: "Toward $\textbf{F}$aithfulness-guided $\textbf{E}$nsemble $\textbf{I}$nterpretation of Neural Network"
_ICLR.cc/2024/Conference — Submitted to ICLR 2024_

### Official Review · Reviewer_8Ctj · 2023-10-31

**Soundness:** 3 good
**Presentation:** 3 good
**Contribution:** 3 good
**Rating:** 6
**Confidence:** 3

**Summary:**

This paper targets interpretable and faithful explanations for specific neural inferences. Currently, feature attribution is a commonly used technique for interpretability and input perturbation is used to objectively quantify the faithfulness of an attribution to the model by masking out salient or monotonous features. These approaches overlook the faithfulness of attribution to the hidden-layer encodings. Thus, this paper tries to measure the faithfulness of hidden layer representations, leading to an optimization method for attribution.

**Strengths:**

Pros:
- This work focuses on an overlooked problem: faithfulness to hidden layer representations, which is worth exploring.
- The ensemble method can remove the need for hyper-parameters and is effective as well.
- Gradient clipping is able to maintain internal faithfulness as shown with visualization results.

**Weaknesses:**

Cons:
- Lack of in-depth discussion of the visualization result. It seems that EP can also obtain good visualization and locate the Submarine better than FEI. It would be better to discuss the impact of 'hot' and 'cold' areas in heat map.

**Questions:**

-

---

> ### Author Response · Authors · 2023-11-14
>
> From the visualizations we presented, certain conclusions can be drawn regarding hot and cold areas. For instance, when the subject is an animal, the model tends to focus more on the animal itself, whereas with objects, only parts of the object may be sufficient for the decision. However, it's important to note that we provided a limited number of visualizations, and generalizing these phenomena might be challenging. Therefore, we refrain from making extensive discussions.
>
> Additionally, we aim to avoid comparing visualizations with segmentation because it assume that the model perceives the image from a segmentation point of view, which might not be true. As you pointed out, EP may appear closer to segmentation for the seaship compared to our method. However, closer resemblance to segmentation does not necessarily imply a better visualization.

---

### Official Review · Reviewer_tHJd · 2023-11-01

**Soundness:** 2 fair
**Presentation:** 2 fair
**Contribution:** 2 fair
**Rating:** 5
**Confidence:** 2

**Summary:**

This paper proposes a novel neural network explanation method driven by faithfulness in both model decisions and internal model functioning. It combines ensemble approximation with gradient clipping. Additionally, it proposes a new qualitative metric that implicitly assesses internal faithfulness. Reasonable qualitative and quantitative results are reported.

**Strengths:**

1. The proposed method makes sense to me.
2. The paper is generally well-written.
3. The qualitative results are reasonable.

**Weaknesses:**

I am not familiar with interpretable deep learning, so I am not sure if the quantitative results are sufficient and significant. The baselines are outdated - Extremal Perturbation(EP) Fong et al. (2019), RISE Petsiuk et al. (2018), FGVis Wagner et al. (2019), GradCam Selvaraju et al. (2017) and Integrated Gradient Sundararajan et al. (2017), so I do not take them as competitors powerful enough. Though the qualitative results with several images make sense to me, I cannot be fully convinced by such an insufficient quantitative comparison.

**Questions:**

I would suggest the authors include quantitative results with recently proposed competitors.

I will lower my rating if the other reviewers' comments on the quality of the quantitative analysis agree with mine.

If the AC and other reviewers are familiar with this research area and believe that the presented results are sufficient enough, please let me know and I will be happy to raise my rating.

---

> ### Author Response · Authors · 2023-11-14
>
> The methods we compared, despite published years ago, still represent state-of-the-art performance. The quantitative evaluation metric we used is  the most commonly employed ones. Unfortunately, recent work has not shown sufficient improvement on these specific metrics. They either only enhance results compared to a specific category of interpretation method (i.e., gradient-based only or perturbation-based only) or propose ad-hoc evaluation metrics themselves. Some older techniques (i.e., grad-cam) presented in the paper actually outperform newer techniques (i.e., extremal perturbation).

---

### Official Review · Reviewer_xnf4 · 2023-11-02

**Soundness:** 2 fair
**Presentation:** 2 fair
**Contribution:** 2 fair
**Rating:** 3
**Confidence:** 2

**Summary:**

The authors propose a new framework for faithfulness based attribution. They use a differentiable approximation of fractiles to allow them to optimize an otherwise non-differentiable objective. They then optimize this and average over different fractiles and use the resulting attribution map to evaluate several metrics.

**Strengths:**

- The paper improves the Q_P metric over the selected baselines.

**Weaknesses:**

- Section 3 is difficult to understand, at least partially due to notation. For example, the LHS of (2) seems to be a scalar based on notation (dot products on the RHS), but from context it seems that the authors are using the \cdot to mean element-wise multiplication (Hadamard product).  There are other things that are confusing like having $\alpha_f(p)$ and then $\alpha_p$, where $p$ presumably means different things in each context - it's an argument, but then it's also used as the fractile. Another example; in (3), $\tilde x$ and $l_{faith}$ should be notated to depend on $f$.

- The relation between faithfulness metrics and faithfulness optimization is unclear

- The results seem mixed at best; there is improvement in Q_P, but overall Q_D is not improved and their defense mechanisms seem to be worse than baseline (this section was difficult to interpret, but based on the text it seems that lower is better).

**Questions:**

1. In 3.2, you define a consistency constraint $\alpha_1 \le \alpha_2$ and then state that optimizing under these constraints is challenging. Have you considered parameterizing $\alpha$ as a cumulative sum? e.g. $\alpha_i = \sum_{j\le i} \delta_j$ In this case, the constraints becomes $\delta_j > 0$ and $\alpha_N=1$, which is easier to handle in an SGD framework.

2. Since there are two faithfulness metrics Q_P and Q_D, how does l_faith relate to them? Is it only optimizing perturbation (a hunch based on table 1 results)?

3. What is $\bar x$ in (5)? Is it the same as $\tilde x$?

4. Are the gradients clipped at every single layer? if not, how do you define the decomposition of the network?

---

> ### Author Response · Authors · 2023-11-14
>
> We have updated the paper, hopefully the updated version can address your concerns and problems.
>
> Notation:
>
> We apologize for any confusion caused by the notation. The dot product in equation (2) should be interpreted as a Hadamard product, and $\alpha_p$ should be written as $\alpha(p)$ in equation (3). The symbol 'p' represents the pixel index of the mapping consistently across these notations. $\bar{x}$ is a typo and should be corrected to $\tilde{x}$. We also change notation for to clarify that $loss_{faith}(f)$ and $\tilde{x}(f)$ corresponding to different fractiles
>
> Faithfulness metric vs faithfulness optimization:
>
> The details of faithfulness metric is described in the faithfulness section in section 2.2. The faithfulness metric can be categorized into two types: deletion metric and preservation metric. The deletion metric assesses the quality of the explanation by systematically removing highly attributed pixels and evaluating the decrease in probability output for the desired category . On the other hand, the preservation metric retains highly attributed pixels while removing others and measures the increase in probability confidence.
>
> Since we don't know how many pixels to operate on, the usual practice is to start with the entire input set to blank and compute the area under the curve (AUC) as $Q_D$, and then proceed from blank to the entire image to compute $Q_P$.
>
> To convert this metric into an optimization problem, a common practice is to use the Hadamard product. For the Preservation metric, we multiply the attribution map with the input and the complement of the map with a reference image. For the Deletion metric, we do the opposite.
>
> However, this approximation lacks knowledge regarding the number of pixels to be operated on. Therefore, we introduce fractiles to approximate a specific point on the curve. Multiple such points are required to interpolate the entire process. Hence, we propose the adoption of an ensemble method involving multiple fractiles.
>
> Faithfulness optimization choice:
>
> In our implementation, we chose optimization formulation for the preservation metric. This choice aligns with our objective of ensuring internal faithfulness in the explanation process, aiming for it to mirror the reasoning process of the original inference and, consequently, produce a similar output. As a result, our evaluation scores tend to favor the preservation metric.
>
> About evaluation:
>
> The quantitative evaluation metric we chose is commonly used. Recent studies typically limit their comparisons to methods within specific categories, like those exclusively gradient-based or perturbation-based. In comparison, our approach conduct a comprehensive comparison across all categories. Gradient-based methods commonly exhibit superior deletion score ($Q_D$) but a terrible visualization quality and preservation score ($Q_P$). Our method, even with a slightly lower performance than the top-performing gradient-based method (Integrated Gradient) in $Q_D$, demonstrates significantly improved preservation scores and visualizations.
>
> Moreover, we also demonstrate the power of gradient clipping in both the defense evaluation and a qualitative visualization.
>
>
> About cumulative sum:
>
> Optimizing the attribution maps as a cumulative sum of $\delta_i$ could offer better consistency guarantees. However, If we  search for all $\delta_i$ at the same time, it is nontrivial to set$ \alpha_N < 1$, so this approach still requires optimizing each \delta_i separately. In this formulation, there is a loss of flexibility and a potential increase in noise. For instance, if $\delta_1=1$, the sum would be fixed at 1, regardless of later $\delta_i$ leading to the accumulation of noise from different $\delta_i$, This issue can be alleviated by increasing the optimization time, but it becomes less efficient. Our cascading optimization procedure, while providing a weaker consistency guarantee, presents a higher chance of eliminating more noise since each map is generated independently.
>
>
> gradient clipping:
>
> Gradient Clipping is applied to the activation layer in the feature extraction part of the neural network as described in end of section 3.3

---

> > ### Comment · Reviewer_xnf4 · 2023-11-23
> >
> > Thank you for addressing many of my comments. I have decided to keep the same score.

---

### Meta-Review · Area_Chair_KgyZ · 2023-12-11

**Metareview:**

This paper introduces a new framework for faithfulness-based attribution, focusing on the interpretability of neural network explanations. The paper received mixed ratings (6, 5, 3, 3). The reviewers acknowledged the improvement in the Q_P metric over selected baselines and its novel approach to hidden-layer encodings, but there are significant concerns about the clarity of Section 3, the relationship between faithfulness metrics and optimization, and the mixed results. While the authors addressed some of these concerns in their rebuttal, the reviewers and AC believe the paper shows promise in the field of interpretability but requires further refinement and clarification. Unfortunately, the paper is not ready for acceptance at this time.

**Justification For Why Not Higher Score:**

The decision is grounded in the reviews and the rebuttal. I believe further refinement and clarification are essential to elevate the paper for acceptance

**Justification For Why Not Lower Score:**

N/A

---

### Decision · Program_Chairs · 2024-01-16

Reject